# Horizontal transfer of probable chicken-pathogenicity chromosomal islands between *Staphylococcus aureus* and *Staphylococcus agnetis*

**Douglas D. Rhoads**[1‡]*, **Jeff Pummil**[1,2‡], **Nnamdi S. Ekesi**[1,3☉], **Adnan A. K. Alrubaye**[1☉]

**1** Program in Cell and Molecular Biology, University of Arkansas, Fayetteville, AR, United States of America, **2** Arkansas High Performance Computing Center, University of Arkansas, Fayetteville, AR, United States of America, **3** Department of Natural Sciences, Northeastern State University, Tahlequah, OK, United States of America

☉ These authors contributed equally to this work.
‡ DDR and JP also contributed equally to this work.
* drhoads@uark.edu

**Data Availability Statement:** All genomes are available in NCBI, and all relevant data are within the paper and its Supporting Information files.

## Abstract

*Staphylococcus agnetis* is an emerging pathogen in chickens but has been most commonly isolated from sub-clinical mastitis in bovines. Previous whole-genome analyses for known virulence genes failed to identify determinants for the switch from mild ductal infections in cattle to severe infections in poultry. We now report identification of a family of 15 kbp, 17–19 gene mobile genetic elements (MGEs) specific to chicken osteomyelitis and dermatitis isolates of *S. agnetis*. These MGEs can be present in multiple copies per genome. The MGE has been vectored on a Staphylococcus phage that separately lysogenized two *S. agnetis* osteomyelitis strains. The *S. agnetis* genome from a broiler breeder case of ulcerative dermatitis contains 2 orthologs of this MGE, not associated with a prophage. BLASTn and phylogenetic analyses show that there are closely related intact MGEs found in genomes of *S. aureus*. The genome from a 1980s isolate from chickens in Ireland contains 3 copies of this MGE. More recent chicken isolates descended from that genome (Poland 2009, Oklahoma 2010, and Arkansas 2018) contain 2 to 4 related copies. Many of the genes of this MGE can be identified in disparate regions of the genomes of other chicken isolates of *S. aureus*. BLAST searches of the NCBI databases detect no similar MGEs outside of *S. aureus* and *S. agnetis*. These MGEs encode no proteins related to those produced by *Staphylococcus aureus* Pathogenicity Islands, which have been associated with the transition of *S. aureus* from human to chicken hosts. Other than mobilization functions, most of the genes in these new MGEs annotate as hypothetical proteins. The MGEs we describe appear to represent a new family of Chromosomal Islands (CIs) shared amongst *S. agnetis* and *S. aureus*. Further work is needed to understand the role of these CIs/MGEs in pathogenesis. Analysis of horizontal transfer of genetic elements between isolates and species of Staphylococci provides clues to evolution of host-pathogen interactions as well as revealing critical determinants for animal welfare and human diseases.

**Funding:** Arkansas Biosciences Institute, and J. William Fulbright College of Arts and Sciences, University of Arkansas. The funders had no role in study design, data collection and analysis, decision to publish, or preparation of the manuscript.

**Competing interests:** The authors have declared that no competing interests exist.

## Introduction

Staphylococcus infections of poultry have been recognized for decades [1–14]. Bacterial chondronecrosis with osteomyelitis (BCO) leading to lameness is a major animal welfare issue in the broiler industry often associated with Staphylococcus infections [8, 9, 14–19]. We published the complete genome of *Staphylococcus agnetis* 908 obtained from BCO lesions in broilers on our poultry research farm [8]. When administered at low dosage ($10^4$ to $10^5$ colony forming units per ml) in drinking water at day 20, this isolate is capable of causing BCO lameness exceeding 50% by day 56 for birds raised under standard conditions [20, 21]. The infection can spread through the air from challenged birds to other birds in the same facility [20, 21]. Our phylogenomic comparisons of *S. agnetis* genomes from chickens and cattle supports the emergence of 908 and other *S. agnetis* infecting chickens from within a wider group of isolates associated with mastitis in cattle [22]. *S. agnetis* is a common isolate from subclinical mastitis in cattle but there are only a few reports of infections in chickens, primarily associated with BCO, septicemia or internal organ colonization [2, 4, 8, 23]. Therefore, *S. agnetis* appears to be an emerging, significant, lethal pathogen in chickens. However, previous whole genome analyses failed to identify any particular genes, or virulence determinants, that distinguish the chicken isolates from the cattle isolates [8, 22], however differences in resident prophage were noted in some of the chicken isolates. More recently we compared the genomes of two sister clades of *S. aureus* where one clade is exclusively isolated from humans and the other from chickens [2]. A previous publication focused on chicken isolates of *S. aureus* from Ireland in the 1980s suggested the switch from humans to chickens was associated with acquisition of mobile genetic elements [19]. Our analyses of a larger collection of genome assemblies for chicken *S. aureus* isolates from the chicken-specific clade supported one specific *Staphylococcus aureus* Pathogenicity Island (SaPI) as key to host specificity within this clade of poultry pathogen genomes [2]. This same SaPI is found in a phylogenetically distant *S. aureus* isolate from China (host unknown). Further, our analyses of additional related genomes of *S. aureus* isolated from chicken hosts from 2010–2019 identified subsequent acquisition of an additional SaPI as this lineage of chicken pathogens continues to evolve [2]. SaPI are mobile genetic elements (MGEs) which can be horizontally transferred through transduction, through co-opting the packaging system of a specific helper phage [24–27]. Insertion often occurs in a specific genomic site [25, 28, 29]. Recently acquired MGEs can often be recognized by distinct codon usage, GC content or GC skew (GC/[G+C]) [30–32]. We therefore sought to re-examine the genomes of chicken and cattle isolates of *S. agnetis* for recent acquisition of MGEs in this emerging chicken pathogen. This identified a second group of MGEs distinct from the SaPI which may also contribute to host or tissue preference in *S. aureus* and *S. agnetis*.

## Methods

Details on genomes utilized for comparisons described in the results are provided in Table 1. Bacterial genome assemblies for *S. aureus* and *S. agnetis* were downloaded for all entries in NCBI using genome_updater v0.5.1 (http://github.com/pirovc). PopPUNK 2.4.0 [33] was utilized to generate Newick trees for genomes, based on kmer comparisons of core and accessory genomes. Sub-genomic regions were extracted using SeqBuilder (LaserGene v 17.3, DNAStar, Madison, WI) and aligned using either Clustal Omega v1.2.4 [34] or the MegalignPro (LaserGene v 17.3) implementation of MAUVE (as indicated). Newick trees were rendered using the MicroReact (https://microreact.org) server [35]. ProkSee Server (https://proksee.ca) was used for visualizing GC skew and BLASTn comparisons of selected bacterial genomes [36]. Genomes were annotated using either the RAST server or prokka v1.14.6 [37]. Local BLASTn and tBLASTn queries used BLAST 2.10.1+ [38, 39]. Prophage sequences were identified using

**Table 1. Genome assemblies utilized in this study.**

| Species | Strain | Accesssion | Host | Reference |
|---|---|---|---|---|
| *S. agnetis* | 908 | GCA_001442815.3 | *Gallus gallus* | [8] |
| *S. agnetis* | 1379 | GCA_011466855.1 | *Bos taurus* | [22] |
| *S. agnetis* | 1416 | GCA_012029465.1 | *G. gallus* | [22] |
| *S. agnetis* | 12B | GCA_007814015.1 | Buffalo | unpublished |
| *S. agnetis* | 722_260714_1_8_heart | GCA_002114335.1 | *G. gallus* | [6] |
| *S. agnetis* | 722_230714_2_5_spleen | GCA_002114365.1 | *G. gallus* | [6] |
| *S. agnetis* | 723_310714_2_2_spleen | GCA_002145465.1 | *G. gallus* | [6] |
| *S. agnetis* | 2044 | JAPTFZ000000000 | *G. gallus* | unpublished |
| *S. aureus* | ED98 | GCA_000024585.1 | *G. gallus* | [19] |
| *S. aureus* | B3-17D | GCA_007726565.1 | *G. gallus* meat | unpublished |
| *S. aureus* | B4-59C | GCA_007726525.1 | *G. gallus* meat | unpublished |
| *S. aureus* | MZ9 | GCA_018622975.1 | Duck | unpublished |
| *S. aureus* | SKY9-1 | GCA_003309755.1 | *Sus scrofa* | unpublished |
| *S. aureus* | YG029 | GCA_003309145.1 | *S. scrofa* | unpublished |
| *S. aureus* | ST398 | GCA_000009585.1 | *Homo sapiens* | [42] |
| *S. aureus* | 16YX14 | GCA_009660995.1 | Pastry | unpublished |
| *S. aureus* | 1AB046 | GCA_003311425.1 | *H. sapiens* | unpublished |
| *S. aureus* | 2009-60-561-1 | GCA_000637255.1 | *G. gallus* | unpublished |
| *S. aureus* | 2011-60-2275-1 | GCA_000637475.1 | *G. gallus* | unpublished |
| *S. aureus* | 2011-60-2275-7 | GCA_000684715.1 | *G. gallus* | unpublished |
| *S. aureus* | 22(2K81-5) | GCA_000684695.1 | *G. gallus* | unpublished |
| *S. aureus* | ch22 | GCA_003350605.1 | *G. gallus* | unpublished |
| *S. aureus* | ch23 | GCA_003336545.1 | *G. gallus* | unpublished |
| *S. aureus* | Chi-10 | GCA_000638915.1 | *G. gallus* | unpublished |
| *S. aureus* | 1516 | GCA_013867445.1 | *G. gallus* | [2] |
| *S. aureus* | K12S0375 | GCA_000934285.1 | *G. gallus* | [43] |
| *E. faecalis* | V583 | GCA_000007785.1 | *H. sapiens* | [44] |

Phaster (https://phaster.ca/) server [40]. Signal peptides were identified using SignalP 6.0 (https://services.healthtech.dtu.dk/service.php?SignalP) server [41].

## Results

To identify recently acquired MGEs based on GC-skew, we used the ProkSee server to compare the main chromosome assembly for chicken isolate *S. agnetis* 908, to the main chromosomes of chicken isolate *S. agnetis* 1416 (Fig 1). The assemblies of the genomes for 908 and 1416 are the most complete assemblies of any chicken *S. agnetis* isolate, with only draft assemblies for three other isolates from Denmark [6]. The Veterinary Disease Research Laboratory at Mississippi State University provided us with 6 suspected *S. agnetis* isolates from a severe case of ulcerative dermatitis from a broiler breeder. Phylogenomic analyses of the assemblies from Illumina sequence data confirmed the six as *S. agnetis* and that these six isolates are clonal (to be published elsewhere). The assembly for isolate 2044 was deemed most complete (fewest contigs with highest assembled bp) and included in the ProkSee analyses (Fig 1). The bovine *S. agnetis* isolates, 1379 and 12B, were included because they represent the most complete assemblies (1379 is a single complete circle; and 12B has the fewest contigs with the highest assembled bp) for bovine *S. agnetis* isolates and our phylogenetic analyses showed them to be closely related to 908 [22]. Based on GC-skew there are two potential MGEs (MGE1 and MGE2)

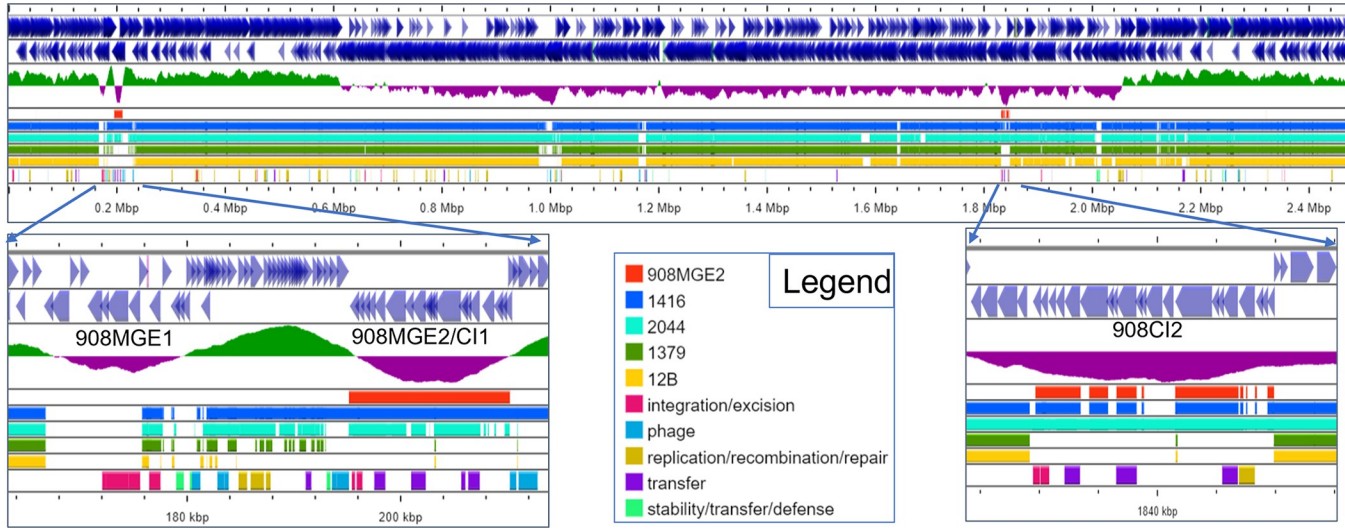

**Fig 1. ProkSee analysis of GC skew and BLASTn to other genomes identifies potential mobile genomic elements in chicken isolates of *Staphylococcus agnetis*.** In Panel A the reference genome was the main chromosome of *S. agnetis* 908 with blue arrows for annotated features (top two rows) followed by the GCskew plot. BLASTn comparisons 908MGE2, chicken isolates 1416 and 2044, and bovine isolates 1379 and 12B are color-coded in the Legend. The last row plots hits to mobile elements from mobileOG-db (see Legend). Panels B and C are expansions of the regions indicated by arrows with mobile elements/ chromosomal islands discussed in the text labeled.

between 0.167 and 0.2102 Mbp in the 908 main chromosome. Only one of these elements, MGE2 near 0.2 Mbp, is also found in the genome of *S. agnetis* 1416, and to a significant extent in *S. agnetis* 2044. Neither potential MGE is found in either of the bovine isolate genome assemblies. Annotation of the *S. agnetis* 908 assembly using the RAST server indicates that MGE1 contains 8 genes spanning 166,820 to 175,593 bp; encoding a three gene beta-lactamase operon, two hypothetical proteins and three proteins related to Tn554 transposases. MGE2 contains 19 genes spanning 195,883–210,152 bp; encoding 10 hypothetical proteins, a conjugal transfer protein, a replication initiation factor, a transposase, a lipoprotein, an acetylglucosa-minidase, a FtsK cell division protein, a AAA family ATPase, and a DUF961 domain protein. Notably, the NCBI Prokaryotic Genome Annotation Pipeline (PGAP) annotation only pre-dicts 17 genes for the MGE2 region. Analysis of the *S. agnetis* 908 genome for probable pro-phage using Phaster shows that 908MGE1 and 908MGE2 are near the boundaries of two prophage; a 46.6 kbp intact Staphylococcus phage EW (accession: NC_007056.1; 45286 bp) from 163,118–209,758 bp, and a 22.4 kbp partial Staphylococcus phage IME-SA4 (accession: NC_029025.1; 41843 bp) from 210,239–232,722 bp. Phaster analysis of the 1416 genome iden-tifies a 49 kb Staphylococcus phage IME-SA4 prophage from 1 to 54210 in a 1.8 Mbp contig. The *S. agnetis* 1416 ortholog of 908MGE2 spans from 17,534–32,604, fully within the prophage in 1416 and not near the prophage boundary as is observed in the 908 genome. In the 1416 genome there is also an incomplete EW prophage from 1,796,052 to the end of the 1.8 Mb 1416 contig. Mauve alignments of the MGE2 homologs from 908 and 1416 demonstrate that the insertion site within the prophage for both isolates are the same (Fig 2). For both prophage the upstream genes are a phage terminase small subunit p27 family gene, and a hypothetical protein, while the downstream genes are two hypothetical proteins followed by a phage portal protein. The evidence is most consistent with the progenitors of isolates 908 and 1416 being separately lysogenized by the same Staphylococcus phage IME-SA4 carrying MGE2 inserted at the same point within the phage genome. However, since the prophage insertion termini are different in 908 and 1416 they represent independent acquisitions in these two *S. agnetis* genomes.

We analyzed the *S. agnetis* 908 genome using Proksee and identified two paralogous regions for 908MGE2 of 15–17 kb (Fig 1). There is the MGE in the prophage and a second region not associated with a prophage. ProkSee BLASTn analyses of the *S. agnetis* 1416 and 2044 genomes queried with 908MGE2 demonstrated that 1416 only has the one copy in the resident prophage, while 2044 contains two copies with neither region associated with a prophage.

We hypothesized that *S. agnetis* 908MGE2 could represent a mobile element related to the switch from cattle to chickens for both 908, 1416, and 2044. To further investigate the source of 908MGE2 we used all 19 predicted polypeptides for a BLASTp of microbial genomes at NCBI. Significant hits (E-value ≤ 0.000001) were only present in genomes of *S. aureus* and *S. agnetis*. For *S. agnetis* the significant hits were specifically the 908 and 1416 genomes we deposited. There were three duck isolate genomes (MZ1, MZ8, MZ9) with significant BLASTp hits that are annotated as *Staphylococcus sp*. However, the similar genome analyses service at BV-BRC supports all three duck isolates being *S. aureus*. Specifically, the best matching genomes for the duck isolates were *S. aureus* subsp. a*ureus* (strains DSM 2031, NCTC 8325, and N315) with distance scores less than 0.009. We downloaded all genome assemblies for *S. aureus* and *S. agnetis* from NCBI and performed a tBLASTn search using the 19 predicted polypeptides of 908MGE2 to identify which genomes contained highly similar coding sequences for the 19 protein encoding genes (PEG218 to PEG236). The most significant hits for each gene in each assembly were tabulated and the tBLASTn results sorted to identify the genomes with the highest average percent identity for the 19 PEGs (S1 Table). Not surprisingly the top hits were for *S. agnetis* 1416 and 908 but highly similar polypeptide products (>87% average protein identity) were found in 11 genomes of *S. aureus*, including 7 isolates from chicken, 2 human isolates, 1 pig isolate, and a food isolate (pastry). There were additional *S. aureus* and *S. agnetis* isolates of lesser average identity (49–70%) which shared some of the 19 PEGs, including 22 from avian sources, 10 from humans, and 10 from other mammals. The tBLASTn searches included the three assemblies of *S. agnetis* from organs of deceased broiler breeders from Denmark: strains 722_230714_2_5_spleen, 722_260714_1_8_heart, and 723_310714_2_2_spleen; [6]. Notably, tBLASTn did not find significantly related sequences to 908MGE2 encoded polypeptides in these draft genome assemblies. Additionally, Phaster analyses of these Danish chicken isolate genomes did detect an intact Staphylococcus phage EW prophage, but no IME-SA4 related prophage. Therefore, the Denmark chicken *S. agnetis* isolates do not appear to have acquired the Staphylococcus prophage carrying MGE2 as identified in *S. agnetis* 908 and 1416. The tBLASTn results (S1 Table) indicate that the 19 PEGs from MGE2 appear to be preferentially found in isolates of *S. aureus* from chickens. We generated a phylogenetic tree based on SNPs in the core genome of the 42 *S. aureus* in S1 Table to determine whether these 19 encoded PEGs were restricted to highly related isolates (Fig 3).

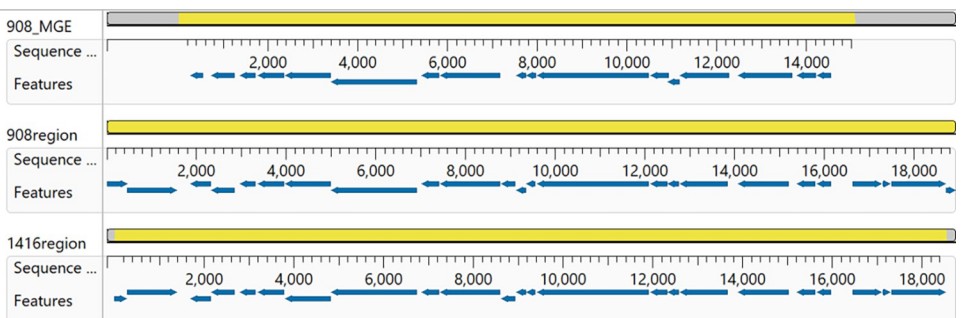

**Fig 2. MAUVE alignment from MegAlign Pro of a mobile element from *Staphylococcus agnetis* 908 (908_MGE) with the region of the insertion from 908 (908region) and *S. agnetis* 1416 (1416region).** For each sequence the features (genes) are displayed as directional arrows for the NCBI annotation of these regions.

Remarkably, the genomes which appear to share a highly related element are not clustered (red nodes). Further, there is no clustering of the avian isolates or the human isolates that share these 19 PEGs.

We next asked if any of the *S. aureus* genomes that share these 19 genes (i.e., YG029, 1AB046, 16YX14, ch22, K12S0375, and ST398) contain a prophage for either Staphyloccus phage IME-SA4 or EW. We did not include *S. aureus* isolates Chi-10, 2009-60-561-1, 2011-60-2275-1, or 2011-60-2275-7, because BLASTn searches with 908MGE2 only identified small contigs (<5000 bp), suggesting that the assemblies for these genomes were of insufficient quality for comparative genomics, while the six genomes surveyed contained the MGE2 ortholog on a much larger contig. Although Phaster did identify between 2 to 7 intact or incomplete phage in the six genomes, none were IME-SA4 or EW. Further, none of those prophages mapped to a region near the most significant BLASTn hit for 908MGE2. Therefore, there is no evidence from these other genomes that 908MGE2 is normally phage associated or that 908MGE2 came from one of these *S. aureus* genomes by transduction of a resident EW or IME-SA4 prophage.

We next investigated whether the 908MGE2 homologs were organized in the same manner in these other genomes. The 908MGE2 sequence was aligned using MAUVE to the assembly contig with the longest, most significant match from the *S. aureus* ST398 and ch22 genome. The alignments showed that MGE2 was co-linear in the ST398 and ch22 *S. aureus* genomes (Fig 4). The ST398 assembly is for a human myocarditis isolate from the ST398 lineage, which is normally associated with live-stock [42]. The NCBI Biosample entry for ch22 indicates this strain was isolated from a deep lesion/wound of a broiler in Poland from 2009. This genome is part of the ST5 lineage and part of a clade that switched hosts from human to chicken prior to 1980, possibly before 1976 [19, 22]. Other closely related genomes from this *S. aureus* chicken-restricted clade include ED98, a 1986 or 1987 isolate from Ireland; ch23 a 2009 isolate from Poland; B3-17D and B4-59C from contaminated chicken meat in 2010 from Tulsa, OK; and 15XX representing 14 clonal assemblies of duplicate isolates from each of seven BCO broilers from a commercial integrator in Northwest Arkansas in 2019 [2, 19]. Surprisingly, the tBLASTn queries indicate some of these genomes, phylogenetically related to ch22, do not contain an element highly colinear to 908MGE2 (S1 Table). The MGE2 region in *S. aureus* ch22 was found to be colinear with respect to *S. agnetis* 908, and 1416, and with *S. aureus* ST398 (Fig 4). Conversely, the genomes for *S. aureus* ED98, B3-17D, and 15XX contain significant (>90% identity) orthologs for some of the PEGs but not for all, specifically lacking orthologs to PEGs 218, 223 and 232 through 235. When we aligned 908MGE2 with these four genome assemblies we discerned that in ED98, B3-17D and 15XX the 908MGE2 orthologs map to 2 or 3 different locations on the main chromosome (S1 Fig). Additionally, alignments of 908MGE2 to the genomes for YG029, 1AB046, 16YX14, K12S0375, SKY9-1, YG029, and ch23, showed that subregions of 908MGE2 map to numerous locations on the chromosome in these *S. aureus* genomes.

SaPI and other phage induced chromosomal islands (PICIs) are generally assumed to have a specific att site for integration [25, 28, 29]. Therefore, we examined the upstream and downstream genes for the three co-linear insertions in *S.aureus* genomes. The upstream and downstream genes are distinct for the insertions in *S. aureus* ch22, ch23, and ST398. In ch22 the upstream gene is purB (adenylosuccinate lyase). In ch23 the upstream genes are three hypothetical proteins. In ST398 the upstream genes are menE (o-succinylbenzoate coA lyase), a DUF4909-containing protein, and a small calcium binding protein (SmCa in Fig 3). The downstream genes in ch22 are scpA (cysteine protease staphopain) and staphostatin A (not marked in Fig 4. In ch23 the downstream genes are three hypothetical proteins. The downstream genes in ST398 are a hypothetical protein, an excalibur calcium-binding domain-

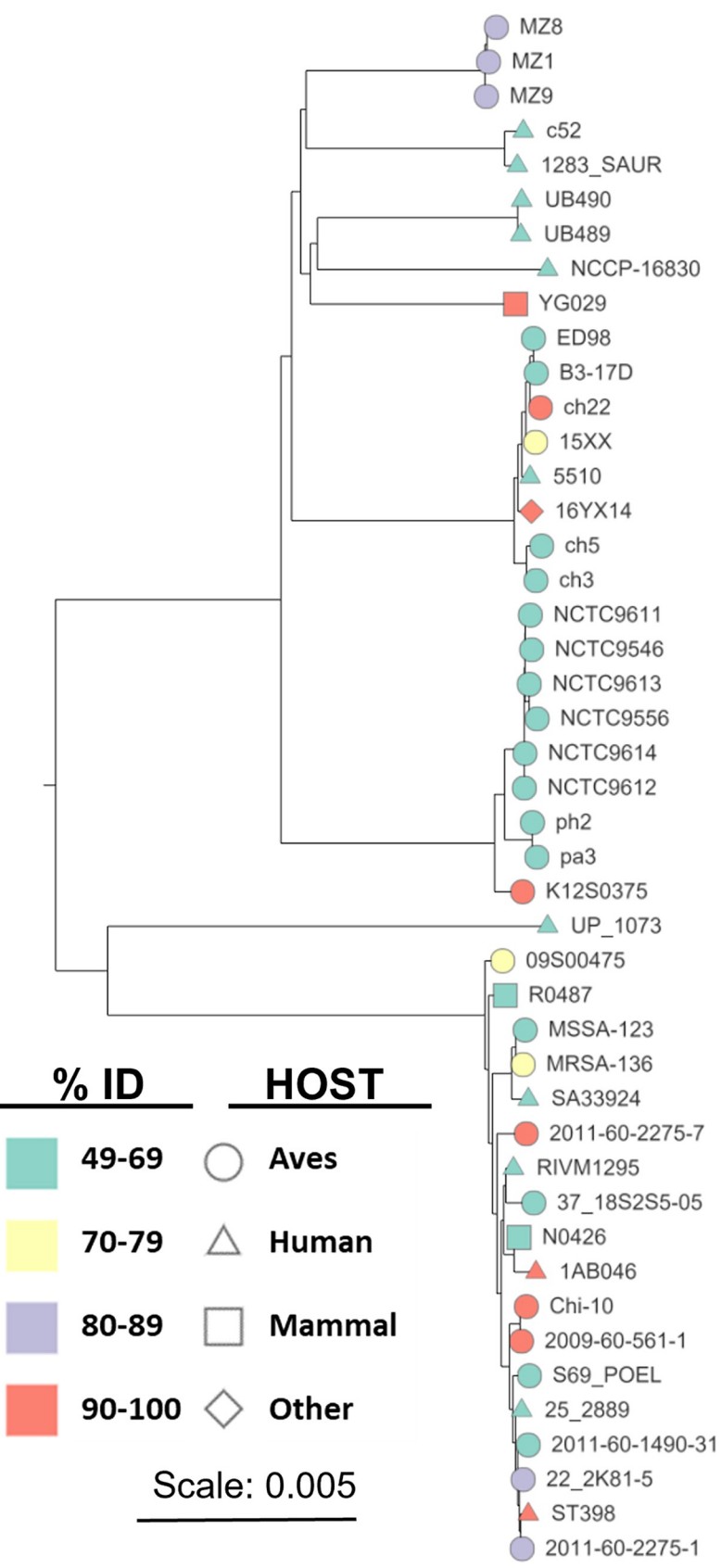

**Fig 3. Phylogenetic tree from poppunk comparison of 42 *Staphylococcus aureus* genomes containing sequences most closely related to a *Staphylococcus agnetis* 908 mobile genetic element.** Nodes are color coded for average tBLASTn percent identity for all 19 genes in the mobile genetic element. Node shape indicates host source. Node labels are strain designations from the NCBI genome database.

containing protein (exCa in Fig 4) and a DUF4352-containing protein. Close inspection of the annotation from PGAP suggests that the exCa coding sequence is partially disrupted suggesting that the insertion of the 908MGE2 ortholog in ST398 may have been within a gene which created the small calcium binding protein upstream and a partial Excalibur Calcium binding protein downstream. Regardless, the insertion sites in ST398, ch22 and ch23 are distinct. Our data find no evidence that 908MGE2 is restricted to a particular location and therefore behaves more like a promiscuous transposon rather than a canonical SaPI/PICI.

As to the possible involvement of the genes in 908MGE2 affecting host-specificity or tissue targeting, there is little evidence of virulence determinants from annotation or BLASTp searches at NCBI (Table 2). We used three annotation pipelines: RAST, PGAP and Bakta, along with BLASTp searches of the NCBI NR. For the RAST annotation all 19 genes annotated as hypothetical, but as can be seen in Table 2, the remaining two pipelines and BLASTp queries, not unsurprisingly, gave often conflicting results. PEGs 218, 224, 229–231, 233, and 236, most likely represent mobilization functions: transposase, FtsK cell division, conjugal transfer, three conjugal transfer proteins, replication/rolling circle initiation factor, and transposon-related protein, respectively. PEGs 223, 225–227, 232, and 234 annotate as hypothetical proteins. PEG 219 and 220 annotate as cystatin-like fold lipoproteins and SignalP predicts that PEG 219 is potentially lipidated. PEG 221 is likely involved in proteoglycan/peptidoglycan cleavage [45]. PEG 222 is predicted by SignalP to contain a signal peptide with permease or nucleic acid binding properties. PEG 228 annotates as an AAA family ATPase which could be involved in energy-dependent conjugal transfer [46]. PEG 235 contains a DUF961 domain (DUF = domain of unknown function). In summary, PEGs 218, 222, 224, 228–231, 233, and

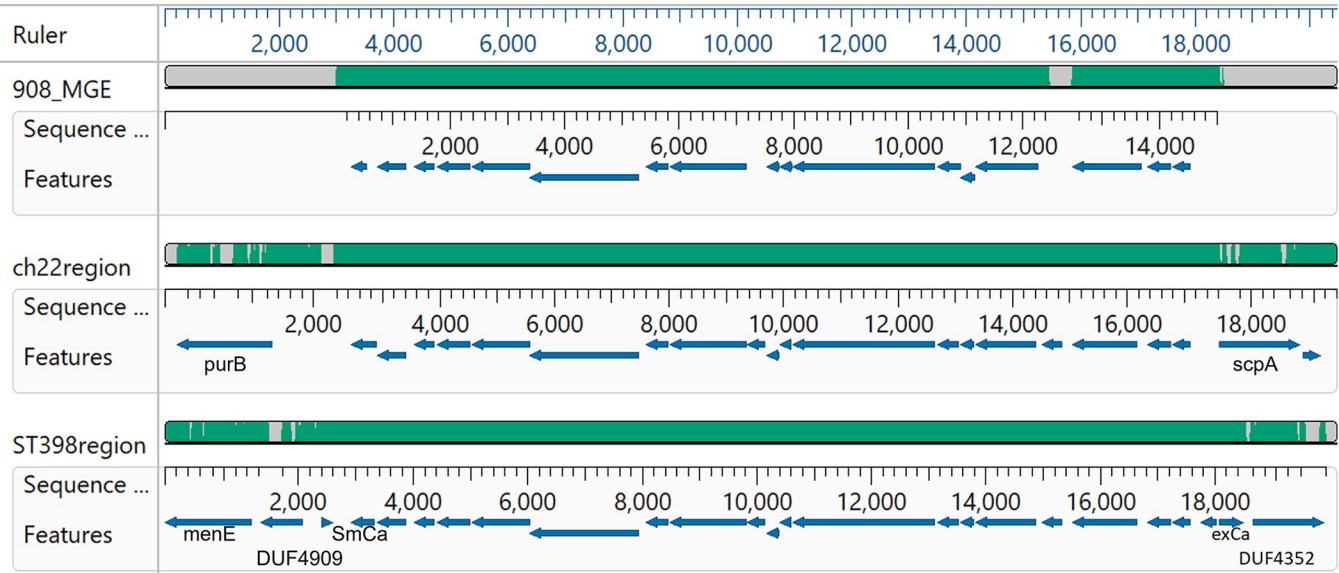

**Fig 4. Clustal omega alignment in MegAlign Pro of the mobile element from *Staphylococcus agnetis* 908 (908_MGE) with the region of the insertion from *Staphylococcus aureus* isolates ST398 (ST398region), ch22 (ch22region), and ch23 (ch23region).** For each sequence the features (genes) are displayed as directional arrows from the NCBI annotation of these regions. Flanking genes discussed in the text are labeled.

**Table 2. Annotation of the 908MGE2 using three different methods.** PEG is protein encoding gene, and AA length is the number of amino acids in the predicted product.

| PEG | AA length | PGAP annotation | BLASTp Annotation | Bakta Annotation |
|---|---|---|---|---|
| 218 | 176 | transposase | helix-turn-helix transposase/integrase | helix-turn-helix domain-containing protein |
| 219 | 120 | cystatin-like fold lipoprotein | cystatin-like fold lipoprotein | DUF4467 domain-containing protein |
| 220 | 197 | hypothetical protein | cystatin-like fold lipoprotein | hypothetical protein |
| 221 | 342 | mannosyl-glycoprotein endo-beta-N-acetylglucosamidase | CHAP domain-containing/glucosaminidase domain/secretory antigen SsaA-like protein | mannosyl-glycoprotein endo-beta-N-acetylglucosamidase |
| 222 | 643 | hypothetical protein | ABC transporter permease | translation initiation factor IF-2 with signal peptide |
| 223 | 133 | hypothetical protein | hypothetical protein | hypothetical protein |
| 224 | 452 | cell division protein FtsK | FtsK/SpoIIIE domain-containing protein | cell division protein FtsK |
| 225 | 110 | hypothetical protein | hypothetical protein | hypothetical protein |
| 226 | 76 | hypothetical protein | hypothetical protein | hypothetical protein |
| 227 | 72 | hypothetical protein | hypothetical protein | hypothetical protein |
| 228 | 831 | AAA family ATPase | ATP-binding protein | AAA family ATPase |
| 229 | 129 | hypothetical protein | TcpE family conjugal transfer membrane protein | conjugal transfer protein |
| 230 | 86 | hypothetical protein | TcpD family membrane protein | hypothetical protein |
| 231 | 358 | conjugal transfer protein | conjugal transfer protein | conjugal transfer protein |
| 232 | 39 | Not annotated | No match | hypothetical protein |
| 233 | 379 | replication initiation factor domain-containing protein | Rolling circle replication initiation factor domain-containing protein | replication initiation factor domain-containing protein |
| 234 | 141 | hypothetical protein | hypothetical protein | hypothetical protein |
| 235 | 107 | DUF961 domain-containing protein | DUF961 family protein | DUF961 domain-containing protein |
| 236 | 94 | hypothetical protein | hypothetical protein | Transposon-related protein |

236, likely encode mobilization and conjugation functions. None of the remaining PEG functions is a particularly strong candidate for a virulence factor, however, some may be extracellular.

Given that 908MGE2 has sequences that map to distinct regions of different genomes from *S. agnetis* and *S. aureus*, we re-examined genomes found to contain 12–16 kb islands identified by BLASTn queries with the 908 MGE-2. Some genomes of *S. agnetis* and *S. aureus* were found to contain 2 or more islands related to 908MGE2 (S1 Fig). Given that this 19 gene element appears to be present in disparate genomes of *S. agnetis* and *S. aureus* isolated primarily from poultry we hypothesized that it represents a family of CIs that are potential PIs. We designated 908MGE2 as Sag908CI1 (Sag for *S. agnetis*). Furthermore, the second, related element from this genome we designated Sag908CI2. Using Sag908CI1 and the ProkSee viewer for BLASTn surveys (S1 Fig), we confirmed that *S. agnetis* 1416 contains only one island, Sag1416CI1, while *S. agnetis* 2044 contains two islands (Sag2044CI1 & Sag2044CI2). The Danish *S. agnetis* chicken isolates (722_260714_1_8_heart, 722_230714_2_5_spleen, 723_310714_2_2_spleen) appear to contain a single island separated onto two contigs in all three assemblies. For *S. aureus* genomes the SKY9-1 genome contains 1 unit-length Sag908CI1 ortholog, whereas *S. aureus* ST398,16YX14 and YG029 contain three. The *S. aureus* 1AB046 genome contains two tandemly arrayed orthologs. For the *S. aureus* ED98-related clade, the number of islands varies with three orthologs of Sag908CI1 in ED98, four in ch22, and only two islands in each of ch23, and B4-59C, even though these genomes cluster within a single clade of the *S. aureus* phylogenomic tree. As mentioned above, there were several genomes with significant tBLASTn matches to 908MGE2/Sag908CI1 (Fig 3) for which we could not

extract a near full length (>14 kbp) ortholog as the tBLASTn and Proksee analyses identified only small contigs (i.e., 2009-60-561-1, 2011-60-2275-1, 2011-60-2275-7, 22(2K81-5), Chi-10, K12S0375). The ProkSee analysis indicates that the reason that our previous tBLASTn surveys showed distributed hits was partly from multiple paralogous islands in many of these genomes.

We generated a phylogenetic tree for 28 representative islands extracted from *S. agnetis* and *S. aureus* genomes (Fig 5). Clustal-Omega alignment was used because it has been shown to be more accurate for sequences with variable termini [47]. For *S. aureus* ED98 the three Sag908CI1- related islands were SauED98CI1, SauED98CI2, and SauED98CI3. The tree in Fig 5 demonstrates that there are, subjectively, at least six lineages for this MGE family, with variations between lineages primarily based on variations upstream of PEG222, within PEG225, upstream of PEG233 and for the cluster of three short PEGs 234–236. The five *S. agnetis* CI were found in 3 lineages with *S. aureus* CI in 5 lineages (Sag2044CI appears to have no close homologs). CI from chicken hosts are present in all 6 lineages, while CI from mammalian hosts are found in 4 lineages. No lineage is restricted to a particular host, although lineage B contains only poultry isolates except for *S. aureus* 16YX14 which is from pastry, which could have come from either contaminated eggs, dairy, or a human worker.

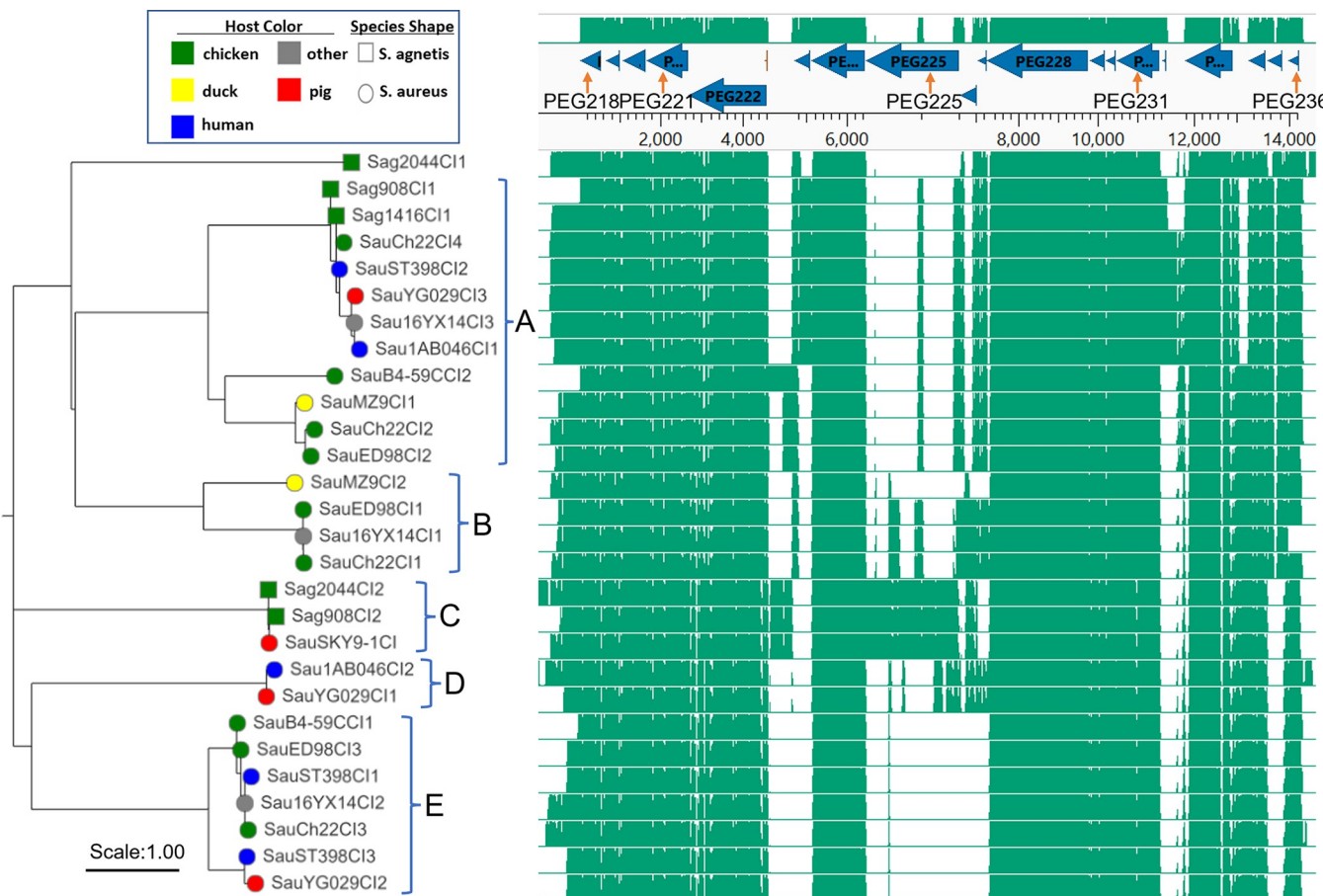

**Fig 5.** Clustal omega alignment (right) and Neighbor-Joining phylogenetic tree (left) for *Staphylococcus aureus* and *Staphylococcus agnetis* chromosomal islands (CI). Bacterial species source is indicated by node shape and host color (legend) and host is coded by label prefix (Sau- *S. aureus*; Sag,—*S. agnetis*. Phylogenetic groups discussed in the text are bracketed on the right of the tree. The location of annotated protein encoding genes (PEG) for 908CI1 are above the alignment with PEG numbers for some genes. Additional explanation of the isolate sources, and numbering are provided in the text.

The two CI we identified from *S. agnetis* 908 are found in lineages A and C. As expected Sag1416CI1 is nearly identical to Sag908CI1 in lineage A, while Sag908CI2 clusters with Sag2044CI2 from the chicken dermatitis isolate. Sag2044CI1 is distant to all the other lineages. ED98 was the earliest (1986 Ireland) genome from the chicken restricted clade of *S. aureus* with ch22 next (2009 Poland), and B4-59C later (2010 Tulsa USA). ED98 contains CIs from lineages A, B, and E. Ch22 contains four CI with 3 closely related to the three from ED98, and the fourth is very closely related to Sag908CI1/Sag1416CI1, which agrees with our earlier tBLASTn surveys of all *S. aureus* genomes. Interestingly, the B4-59C genome contains only two CIs with one closely related to SauED98CI3 and SauCh22CI3 in lineage E, while SauB4-59CC12, is in lineage A and appears to be diverged from the lineage A CIs from ED98 and ch22. Thus, the elements appear to be evolving at a significant rate or there was a loss of 2 and gain of a new CI in the generation of *S. aureus* B4-59C. Indeed, inspection of these three genomes using ProkSee server shows that ED98 and ch22 share CI in three locations with ch22 acquiring an additional element approximately 50 kbp distal to SauCh22CI3 (S1 Fig). SauB4-59CI1 shares the same location as SauED98CI3/SauCh22CI3, consistent with the phylogenetic reconstruction. B4-59C lacks or has lost the elements corresponding to SauED98CI1/SauCh22CI1, and SauED98CI2/SauCh22CI2, with SauB4-59CCI2 inserted 600 kb distal to SauB4-59CCI1. Based on the sequence of SauB4-59CCI2 this appears to be a newly acquired CI and not a transposition of one of the pre-existing elements.

To more precisely map the boundaries of representative CI in both *S. agnetis* and *S. aureus* we identified two instances where there is a CI in a genome where there is a closely related genome that lacks that particular CI. This was accomplished using the phylogenetic relationships of the CI (Fig 5) and previously published whole genome relationships within each species [2, 22]. For *S. agnetis*, the chicken isolate 908 contains two CI insertions and the cattle-isolate 1379 lacks any CI insertions. The insertion site for Sag908CI1 is within a prophage (which is absent from the 1379 genome). However, MAFFT alignments of the 908 and 1379 genomes clearly reveal the insertion site for Sag908CI2 (Fig 6) and the corresponding region in *S. agnetis* 1379 lacks the CI insertion. For *S. aureus* we similarly identified an insertion that distinguishes the genomes for ED98 and B4-59C. As described above, ED98 contains three CI insertions where B4-59C shares two of these and a third CI not related to the third element of

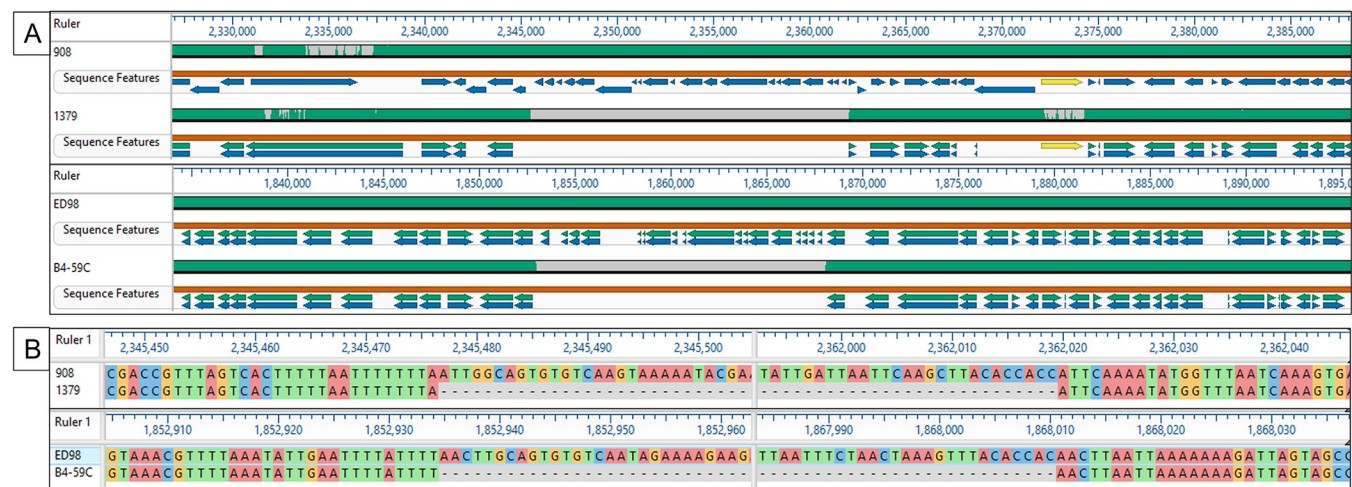

**Fig 6. MAFFT alignments for genomic regions with or without chromosomal island insertions in *Staphylococcus agnetis* (908 vs 1379) and *Staphylococcus aureus* (ED98 vs B4-59C).** Panel A is the encompassing regions with arrows depicting the NCBI gene annotations, and demonstrating conservation of the paired genomes. Panel B shows the aligned termini and proximal flanking regions for each chromosomal island insertion.

ED98. The genome of *S. aureus* B4-59C lacks an insertion corresponding to SauED98CI1. Note that we cannot distinguish whether for either 908 vs 1379, or ED98 vs B4-59C the CI absence is because there was no insertion, or whether there was excision. In both cases, Fig 6A shows that the annotated genes surrounding the insertion site are highly concordant with some minor differences in the annotated genes. Fig 6B delineates the DNA sequences of the CI termini for each insert, and neither appears to include any direct or inverted repeats, which would be characteristic of many transposons [48] or insertion sequences [49]. The flanking sequences also differ between the two insertion sites, but both insertions appear to be in AT-rich regions.

We then asked if the CI we have described represent a distant lineage related to known CI from *S. aureus* or other Gram[+] cocci. We included SaPI sequences from human *S. aureus* isolates OC3, RN3984, COL, and JP5338, representing the archetypal SaPI1, SaPI2, SaPI3, and SaPIbov5, respectively [24, 28, 29, 50, 51]. We included distantly related PICIs from *Enterococcus faecalis* (EfCIV583) and from *Lactobacillus lactis* (LlCIbIL310). These latter two CI have been shown to share structural similarities to SaPIs including a bi-directional promoter and repressor/replicase orientation [52–54]. We also included the SaPI from ED98 which has been previously described [2, 19]. We queried these five previously recognized SaPI, and the two CI representing families from other Gram[+] bacteria by tBLASTn with the predicted 19 polypeptides from Sag908CI1. The best match was one 21 amino acid sequence in PEG222 with SaPI-bov5 with 47% identity, 6% query coverage, and 0.016 Evalue. All other "matches had Evalue > 1. Therefore, there appear to be no shared polypeptide motifs. The SaPI and PICI are mobilized by helper phage [25, 28, 29, 50, 55]. In addition, they are all characterized as having a bidirectional promoter involved in transcription repression [25, 29, 50, 53, 56]. The original discovery and description of SaPI was that they encoded toxin genes. Annotation of Sag908CI1 as well as others from lineages B, C and E fail to identify any recognizable toxin genes. However, even the SaPI in ED98 harbors no recognizable virulence determinants [2, 19]. In addition, we examined the annotations for Sag908CI1 and other examples from lineage B, D and E (e.g., Sauch22CI1, Sauch22CI2, Sauch22CI3, Sauch22CI4, Sag908CI2) and all show only unidirectional transcription (for examples see Figs 2 and 4). Further, the annotation seems to suggest that these CI are mobilized by conjugation (Table 2). However, the genomic evidence from Sag908CI1 and SAg1461CI1 demonstrate transfer to *S. agnetis* from *S. aureus* can involve transduction as part of a Staphylococcal prophage. However, that is clearly not the only mechanism for horizontal transfer since Sag980CI2, Sag2044CI2 are closely related to SauSKY9-1CI1 in a pig isolate from *S. aureus*, the evidence suggests that horizontal transmission can occur via other mechanisms, perhaps conjugation. The SaPI in ED98 and its relatives has been correlated with a host switch by *S. aureus* from human to chickens [19]. This same SaPI is found in all descendants of this chicken-restricted clade [2, 19], and recent genomes from this clade have been found to have an additional SaPI [2]. There is no evidence of acquisition of a SaPI in the transition of *S. agnetis* from cattle to chickens, but we have identified the acquisition of a different CI that correlates with this host switch and this CI and paralogs are found in *S. aureus* pathogenizing chickens including those from the ED98 clade.

## Discussion

Staphylococcal species are known to colonize and cause opportunistic disease in a variety of vertebrates [57–60]. Particular species and sequence-types are associated with particular hosts or niches within their host. *S. aureus* has been associated with severe human infections of skin, bone, and respiratory tract, which can be especially problematic when the infection involves a drug resistant strain [58, 61–63]. A number of *S. aureus* virulence determinants have been

identified that contribute to disease severity [27, 62, 64–69], but little is known about the genetics of host or niche specificity. Host specificity in *S. aureus* has been correlated with variations in host protein binding factors [24, 70], or toxin production [1, 58]. Horizontal transfer of a MGE has been correlated with the switch from human to chickens in one clade of *S. aureus* in the 1980s [2, 19, 27]. The switch correlates with acquisition of a SaPI but the genes in that MGE are mostly un-annotated or hypothetical, so the contributions of that SaPI to colonization or disease of poultry are unknown [2]. The genomes of isolates from this chicken-restricted clade continue to evolve as isolates from 2010 and 2019 have an additional SaPI carrying a toxic shock syndrome toxin 1 and a exotoxin superantigen [2].

We first identified *S. agnetis* as the predominant bacterial isolate from chicken BCO lesions on our research farm [8]. Genome sequence of multiple isolates suggested a largely clonal population with *S. agnetis* 908 representative. We suspect that this hypervirulent pathogen resulted from many years of inducing high levels of BCO within the same facility [20, 21, 23]. *S. agnetis* had previously only been associated with sub-clinical mastitis in cattle [71–73]. More recently we and others have reported additional isolates of *S. agnetis* infecting chickens [4, 6, 22]. Examples include *S. agnetis* 1416 from a BCO lesion, and 2044 from ulcerative dermatitis. Phylogenomic analyses demonstrate that the chicken isolates are highly related to the cattle mastitis isolates and the chicken isolates appear to arise from within the cattle isolates [22]. However, the switch from cattle to chickens is more than just a change in host-specificity. The infection in cattle appears to be restricted to the epithelia and ducts of the mammary gland, whereas in chickens the infection often targets organs, blood, and bone [4, 6, 8, 15, 23, 71, 74]. *S. agnetis* 908 and a human BCO *S. aureus* isolate have been shown to induce cell death in osteoblasts through an NLRP3 inflammasome reaction [75]. However, our initial comparisons of genome assemblies from cattle and chicken isolates failed to identify any specific loss or gain of virulence genes associated with the switch of host and niche [8, 22]. Based on the association of host switch in *S. aureus* with acquisition of a SaPI, we revisited the genome analyses of cattle and chicken isolates of *S. agnetis* with a focus on GCskew to identify recent acquisitions. This identified a 17 to 19 gene region we designated 908MGE2 spanning approximately 15 kb within a Staphylococcus prophage. This prophage was separately acquired by two distinct chicken *S. agnetis* isolates, 908 and 1416. These two isolates were obtained from BCO lesions from different facilities in Arkansas and the insertions of the prophage are different. Searches of the *S. aureus* and *S. agnetis* genomes in NCBI identified 11 *S. aureus* genomes with highly related genes to all of the genes in 908MGE2. Six of these isolates were obtained from chicken, and there is a 7[th] isolate from pastry which could derive from contaminated eggs. There are three draft *S. agnetis* genome assemblies for isolates obtained from organs of deceased broilers in Denmark [6]. These three genomes represent a larger set of *S. agnetis* isolates obtained from diseased broiler breeders and from cloacal samples of newborn chicks. BLASTn analyses suggest that sequences related to 908MGE2 are found in the Danish isolate genomes but separated on two small contigs. *S. agnetis* 2044 was found to contain two regions related to 908MGE2, and the 908 genome actually contains two distinct copies of this MGE. Therefore, if 908MGE2 and its paralog are related to the switch from cattle to chickens then this MGE may be sufficient for chicken pathogenesis by *S. agnetis*, as different chicken isolates contain related representatives of this element. However, there remain only 6 genome assemblies for chicken isolates of *S. agnetis* in the genome databases. Only three *S. aureus* assemblies (ST398, ch22 and ch23) were found to have 908MGE2 with the same gene arrangement as in *S. agnetis* 908 and 1416. ST398 is a human endocarditis isolate of the ST398 lineage which has been primarily associated with livestock. Ch22 and ch23 are from the ST5 lineage and were obtained from deep wounds/lesions from chickens in Poland in 2008. Ch22 and ch23 are closely related members of the chicken-restricted clade of *S. aureus*; obtained from chickens in the United

Kingdom, Poland, and the USA [2, 19]. Isolates of this lineage have been obtained from contaminated chicken meat in Oklahoma, and from BCO lesions in broilers in a commercial production facility in Northwest Arkansas. However, 908MGE2 does not appear to be intact in some members of the chicken-restricted clade. Several of the predicted polypeptides are not found, while others are conserved but portions of 908MGE2 are found in distinct locations on the main chromosome. Since this intact MGE is found in distinct isolates of *S. aureus* and *S. agnetis* we have investigated whether it constitutes a new class of CI distinct from the SaPI of *S. aureus* and PICI of other Gram⁺ cocci. Indeed, phylogenetic analyses (Fig 5) demonstrate that there are five different lineages of approximately 15 kb MGE/CI in *S.aureus*/*S. agnetis* and another lineage with only one representative from *S. agnetis*. These new lineages of CI do not appear to be classic SaPI or PICI, but may be conjugative transposons, based on limited annotation information.

S. agnetis is an emerging pathogen in chickens [4, 6, 8]. Our work has identified app. 15 kb CIs in genomes of *S. agnetis* from severe disease incidences in chickens where these islands are highly related to DNA islands found in genomes of pathogenic *S. aureus*. Indeed, in the evolution of the ED98 related clade of *S. aureus* there is evidence of loss and acquisition of the members of this CI family. These islands are not related to the well characterized MGEs known as SaPI, as SaPI are restricted to particular att sites [25, 29, 51, 54, 55]. The islands we have identified do not appear to be restricted to a particular att site (Fig 4), and the islands shares no significant sequence homology with SaPI. The coordinate acquisition of these elements with transition to chicken pathogenesis for *S. agnetis* and their affiliation with known *S. aureus* pathogens of chickens, supports these elements as CI and possibly as PI. If the genes of 908MGE2 acquired by *S. agnetis* 908 and 1416 are critical for host or niche specificity, then it may be possible to focus on only those conserved across all the representatives of the lineage from a particular host. Only two of the genes in Sag908CI1 are predicted to be externalized, PEG 219 and 222. *S. agnetis* 908 has been shown to trigger an NLRP3 inflammasome response in osteoblast cultures [75]. Femoral head necrosis, one form of BCO, has been correlated with down-regulation of autophagy, and isolate 908 decreases viability of, and expression of autophagy genes of, human fetal osteoblast cells [76]. We have found that *S. agnetis* 908 survives in and escapes from chicken immortalized macrophage-like cells [77]. It should be possible to engineer *S. agnetis* 908 and/or 1416 to remove the entire CI or selected genes, to test whether these responses in tissue culture are affected. This would be advantageous relative to testing in whole animal experiments, which would require biosecurity containment facilities not readily available. Further work is needed to dissect the key genetics of host-switching and niche-targeting in the Staphylococci. This further work would expand investigations on horizontal transfer of MGE as key to bacterial pathogen dynamics [24, 25, 27, 58, 78]. MGE have been associated with the emergence of *Yersinia pestis* and *Vibrio cholerae* from non-pathogenic commensals in humans [79]. Our data is most consistent with 908MGE2/Sag908CI1 being acquired by a Staphylococcus phage from the genome of a *S. aureus* highly-related to ch22 and ch23, infecting chickens. That phage then lysogenized *S. agnetis* separately to generate isolates 908 and 1416 possibly contributing to host switch from cattle to chickens. This scenario is further supported in that cattle and chickens are often raised in close proximity, both *S. aureus* and *S. agnetis* are commonly isolated from cattle, and we have shown that *S. agnetis* infections can be spread through the air [20, 21]. The chicken affords an excellent model for some of these studies as there are clear examples from *S. agnetis* and *S. aureus* of host restriction [2, 19]. Further research could not only reduce an economically important animal welfare issue (i.e., BCO-lameness) but also elucidate mechanisms critical to human and zoonotic Staphylococcal infections.

## Supporting information

**S1 Fig. Proksee views of Staphylococcus genomes containing orthologs of 908MGE2.** For each panel the strain is indicated as in Fig 5 with NCBI annotated genes as blue arrows at the top, followed by GCskew (green-magenta plot), and 908MGE2 blastn hits in red.
(TIF)

**S1 Table. Percent identity scores from a tBLASTn of *S. agnetis* and *S.aureus* genomes from NCBI when queried with PEG218-236 from the *S. agnetis* 908MGE2.** For each genome the strain, species, and host/source, are based on the Biosample entry in NCBI. Genomes were sorted based on average percent identify for all 19 genes, and cells color coded (Green to Red) for percent identity using Microsoft Excel conditional formatting.
(DOCX)

## Acknowledgments

Dr. Mark Hart provided thoughtful comments regarding this manuscript.

## Author Contributions

**Conceptualization:** Douglas D. Rhoads.

**Formal analysis:** Douglas D. Rhoads, Jeff Pummil, Nnamdi S. Ekesi.

**Funding acquisition:** Adnan A. K. Alrubaye.

**Investigation:** Douglas D. Rhoads, Jeff Pummil, Nnamdi S. Ekesi, Adnan A. K. Alrubaye.

**Methodology:** Douglas D. Rhoads, Jeff Pummil.

**Project administration:** Douglas D. Rhoads.

**Resources:** Douglas D. Rhoads, Adnan A. K. Alrubaye.

**Supervision:** Douglas D. Rhoads.

**Writing – original draft:** Douglas D. Rhoads.

**Writing – review & editing:** Jeff Pummil, Nnamdi S. Ekesi, Adnan A. K. Alrubaye.

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
