## [Decision Letter · Decision Letter 0]

12 May 2023

PONE-D-23-08188Horizontal transfer of probable chicken-pathogenicity chromosomal islands between Staphylococcus aureus and Staphylococcus agnetisPLOS ONE

Dear Dr. Rhoads,

Thank you for submitting your manuscript to PLOS ONE. After careful consideration, we feel that it has merit but does not fully meet PLOS ONE’s publication criteria as it currently stands. Therefore, we invite you to submit a revised version of the manuscript that addresses the points raised during the review process.

We look forward to receiving your revised manuscript.

Kind regards,

Feng Gao

Academic Editor

PLOS ONE

Journal Requirements:

5. We note that you have referenced (unpublished) on page 22 which has currently not yet been accepted for publication. Please remove this from your References and amend this to state in the body of your manuscript: (ie “Bewick et al. [Unpublished]”) as detailed online in our guide for authors

6. Please upload a copy of Supporting Information Figure/Table/etc. Supplementary Table 1 which you refer to in your text on page 9.

Reviewers' comments:

Reviewer's Responses to Questions

**Comments to the Author**

1. Is the manuscript technically sound, and do the data support the conclusions?

Reviewer #1: Yes

Reviewer #2: Yes

2. Has the statistical analysis been performed appropriately and rigorously? 

Reviewer #1: Yes

Reviewer #2: Yes

3. Have the authors made all data underlying the findings in their manuscript fully available?

Reviewer #1: No

Reviewer #2: No

4. Is the manuscript presented in an intelligible fashion and written in standard English?

Reviewer #1: Yes

Reviewer #2: Yes

5. Review Comments to the Author

Reviewer #1: The article reports the discovery of a mobile genetic element (MGE) in Staphylococcus agnetis isolates from chicken osteomyelitis and dermatitis.This MGE is only found in chicken infections and can have multiple copies per genome.

It was vectored on a Staphylococcus phage and is related to intact MGEs found in S. aureus genomes.

According to the article, these MGEs may represent a new family of Chromosomal Islands shared by S. agnetis and S. aureus, and more research is needed to understand their role in pathogenesis.

I have a couple of minor remarks:

1-It would be great if the authors could include the species names in Figures 1–4 as they did in Figure 5.

Without the species names, there are too many strain names, which can be confusing.

2-In Fig 2, at line 188, the authors mention dark green nodes.

Did they mean dark blue nodes, or are they referring to fig 3?

3-From time to time, authors discuss data that lack figures (data not shown). Because these data complete the picture, I believe some of these data should be included in the supplementary figures.

I am excited to learn more about how these MGEs contribute to pathogenesis.

Reviewer #2: 1. Manuscript switches between referring to an isolate with their numerical name e.g. line 101 and isolate "numerical name" e.g. line 107. Consistency is recommended. It is easier to follow the manuscript when isolates are referred as "isolate numerical name".

2. Lines 107 and 109 refer to genomes as "most complete" but lack descriptive numbers to determine a genome as close to complete.

3. Line 111 verbiage "from 0.16 to just beyond 0.2 Mbp" is vague. Suggestion for replacement: "in the region between 0.16 - 0.24 Mbp".

4. Lines 131 and 245-246 mention insertion site for the MGEs. What are the nucleotide sequences of the insertion sites in the genomes mentioned? What are the bp numbers for the insertion site in the isolate 908 genome? With the lack of insertion site sequence, there is not enough evidence provided to support the claim that the insertion sites are distinct. Authors have compared annotations of the regions around the MGE but not supported that with nucleotide level identities of the surrounding regions.

5. What are the direct or indirect repeats associated with 908 MGE1, 908MGE2/CI1 and 908CI2?

6. PCR is needed to the presence of the MGEs at the insertion region as determined by the genome assembly is correct.

7. Line 165 mentions use of genome analyses service at BV-BRC to support all three duck isolates being S. aureus. Manuscript lacks the parameters used to conclude duck isolates as S. aureus.

8. Supplementary Table 1 mentioned in line 171 is missing.

9. Fig 2 referred in line 187 does not sow a phylogenetic tree. Did the authors mean Fig 4?

10. Data referred to in lines 219 - 222 should be included as supplementary figure.

11. Table 2 legend: sentence beginning with "PEG is protein..." does not need to be bold.

12. Lines 281-286 need to be supported by data. Data can be added to the supplementary materials.

13. Line 303: should be a period instead of comma after "SauED98CI3".

6. PLOS authors have the option to publish the peer review history of their article (what does this mean?). If published, this will include your full peer review and any attached files.

Reviewer #1: No

Reviewer #2: No

---

## [Author Response · Author response to Decision Letter 0]

1 Jun 2023

Response: none required

Response: None of the funding for this project had grant numbers. They were all internal awards.

Response: we apologize for not including the Supplementary Table 1 in our original submission. The table is now uploaded with the manuscript.

Response: all occurrences of “data not shown” have been dealt with either removed or replaced with supplementary tables/figures.

5. We note that you have referenced (unpublished) on page 22 which has currently not yet been accepted for publication. Please remove this from your References and amend this to state in the body of your manuscript: (ie “Bewick et al. [Unpublished]”) as detailed online in our guide for authors

Response: The styles link documents what to do for “Accepted, unpublished articles”. At this time we have no plans to publish the information listed regarding S. agnetis 908 survival in macrophage-like cells, and it will likely remain “unpublished”. Further, many of the genomes from NCBI (Table 1) were submitted but never published. We await editor suggestions on Table 1, and have modified the text (lines 485-487) 

6. Please upload a copy of Supporting Information Figure/Table/etc. Supplementary Table 1 which you refer to in your text on page 9.

Response: this has been done

Response: not applicable

Reviewers' comments:

Reviewer's Responses to Questions

Comments to the Author

1. Is the manuscript technically sound, and do the data support the conclusions?

Reviewer #1: Yes

Reviewer #2: Yes

2. Has the statistical analysis been performed appropriately and rigorously? 

Reviewer #1: Yes

Reviewer #2: Yes

3. Have the authors made all data underlying the findings in their manuscript fully available?

Reviewer #1: No

Reviewer #2: No

4. Is the manuscript presented in an intelligible fashion and written in standard English?

Reviewer #1: Yes

Reviewer #2: Yes

5. Review Comments to the Author

Reviewer #1: The article reports the discovery of a mobile genetic element (MGE) in Staphylococcus agnetis isolates from chicken osteomyelitis and dermatitis.This MGE is only found in chicken infections and can have multiple copies per genome.

It was vectored on a Staphylococcus phage and is related to intact MGEs found in S. aureus genomes.

According to the article, these MGEs may represent a new family of Chromosomal Islands shared by S. agnetis and S. aureus, and more research is needed to understand their role in pathogenesis.

I have a couple of minor remarks:

1-It would be great if the authors could include the species names in Figures 1–4 as they did in Figure 5.

Without the species names, there are too many strain names, which can be confusing.

Response: Only Figure 5 includes multiple isolates of both strains so we included the genus-species abbreviated prefix there for the phylogenomic tree clarity. In the other figures the species are clearly defined in the legend, especially in figure 4 where only S. aureus isolates are in the tree. In figures 1 the legend states that it is only S. agnetis in the figure, Figure 2 is only two isolates of S. agnetis (made clearer in the revision), and in Figure 3 we clarified the species in the legend. Although the reviewers point is well taken that it is likely difficult to most to keep all the “players” clear. We have modified the manuscript to repeatedly specify the species so that it is easier to follow which species each isolate is. 

2-In Fig 2, at line 188, the authors mention dark green nodes.

Did they mean dark blue nodes, or are they referring to fig 3?

Response: This is definitely our error in tracking figures during revisions. This should refer to the poppunk tree in our submitted Figure 4, and the nodes we are referring to are actually in red (as we rendered the tree through multiple iterations). To correct our mistake we switched Figure 3 and 4 in the revision. Figure citations have therefore been updated.

3-From time to time, authors discuss data that lack figures (data not shown). Because these data complete the picture, I believe some of these data should be included in the supplementary figures.

Response: We have added Figure 6 and S Figure 1 to provide additional data and remove the “data not shown”

I am excited to learn more about how these MGEs contribute to pathogenesis.

Reviewer #2: 1. Manuscript switches between referring to an isolate with their numerical name e.g. line 101 and isolate "numerical name" e.g. line 107. Consistency is recommended. It is easier to follow the manuscript when isolates are referred as "isolate numerical name".

Response: we have endeavored to do so throughout. We have modified the manuscript to repeatedly specify the species so that it is easier to follow which species each isolate is.

2. Lines 107 and 109 refer to genomes as "most complete" but lack descriptive numbers to determine a genome as close to complete.

Response: we have added the criteria for our selections. (lines 107-111)

3. Line 111 verbiage "from 0.16 to just beyond 0.2 Mbp" is vague. Suggestion for replacement: " in the region between 0.16 - 0.24 Mbp ".

Response: we have done so. (lines 112-114)

4. Lines 131 and 245-246 mention insertion site for the MGEs. What are the nucleotide sequences of the insertion sites in the genomes mentioned? What are the bp numbers for the insertion site in the isolate 908 genome? With the lack of insertion site sequence, there is not enough evidence provided to support the claim that the insertion sites are distinct. Authors have compared annotations of the regions around the MGE but not supported that with nucleotide level identities of the surrounding regions.

Response: we had not tried to identify the specific termini of the MGE because to know the insertion site you must have the prophage with and without the MGE. We had looked for direct repeats but did not describe this. Based on the reviewers comments we have reanalyzed genomes where there are the best candidates for sequences before and after insertion which has led to inclusion of Figure 6 where we analyzed the insertion site in S. agnetis for 908CI2 because we have a closely related cattle-isolate 1379, and then in S. aureus where ED98 has 3 insertions but the the closely related isolate B4-59C only has 2, so we can align the genomes for both these cases to specifically identify the island termini. We have therefore expanded the discussion about insertion site, and included Figure 6. We thank the reviewer for the comments as the new data and figure provide necessary details.

5. What are the direct or indirect repeats associated with 908 MGE1, 908MGE2/CI1 and 908CI2?

Response: see preceding response

6. PCR is needed to the presence of the MGEs at the insertion region as determined by the genome assembly is correct.

Response: it is not clear what the reviewer is requesting. Are they requesting that we verify that the assembly is correct through some sort of PCR test? Given that the focus is really on a family of chromosomal islands in multiple genomes and many of those genomes were assembled by others we could only have access to DNA from a few genomes. 

7. Line 165 mentions use of genome analyses service at BV-BRC to support all three duck isolates being S. aureus. Manuscript lacks the parameters used to conclude duck isolates as S. aureus.

Response: we have included the specific metric (lines 167-169)

8. Supplementary Table 1 mentioned in line 171 is missing.

Response: We deeply apologize for this error on our part. The table has now been included

9. Fig 2 referred in line 187 does not sow a phylogenetic tree. Did the authors mean Fig 4?

Response: This error has been corrected and includes the switching of Figure 3 and 4.

10. Data referred to in lines 219 - 222 should be included as supplementary figure.

Response: We have done so.

11. Table 2 legend: sentence beginning with "PEG is protein..." does not need to be bold.

Response: corrected.

12. Lines 281-286 need to be supported by data. Data can be added to the supplementary materials.

Response: we have done so with S1 Figure.

13. Line 303: should be a period instead of comma after "SauED98CI3".

Response: Corrected (line 307)

---

## [Decision Letter · Decision Letter 1]

22 Jun 2023

Horizontal transfer of probable chicken-pathogenicity chromosomal islands between Staphylococcus aureus and Staphylococcus agnetis

PONE-D-23-08188R1

Dear Dr. Rhoads,

We’re pleased to inform you that your manuscript has been judged scientifically suitable for publication and will be formally accepted for publication once it meets all outstanding technical requirements.

Kind regards,

Feng Gao

Academic Editor

PLOS ONE

Additional Editor Comments (optional):

Reviewers' comments:

Reviewer's Responses to Questions

**Comments to the Author**

1. If the authors have adequately addressed your comments raised in a previous round of review and you feel that this manuscript is now acceptable for publication, you may indicate that here to bypass the “Comments to the Author” section, enter your conflict of interest statement in the “Confidential to Editor” section, and submit your "Accept" recommendation.

Reviewer #1: All comments have been addressed

Reviewer #2: All comments have been addressed

2. Is the manuscript technically sound, and do the data support the conclusions?

Reviewer #1: (No Response)

Reviewer #2: Yes

3. Has the statistical analysis been performed appropriately and rigorously? 

Reviewer #1: Yes

Reviewer #2: Yes

4. Have the authors made all data underlying the findings in their manuscript fully available?

Reviewer #1: Yes

Reviewer #2: Yes

5. Is the manuscript presented in an intelligible fashion and written in standard English?

Reviewer #1: Yes

Reviewer #2: Yes

6. Review Comments to the Author

Reviewer #1: Dear Authors,

Thank you so much for taking my comments into account and modifying the manuscript accordingly.

Reviewer #2: (No Response)

7. PLOS authors have the option to publish the peer review history of their article (what does this mean?). If published, this will include your full peer review and any attached files.

Reviewer #1: No

Reviewer #2: No

---

## [Editor Report · Acceptance letter]

26 Jun 2023

PONE-D-23-08188R1 

Horizontal transfer of probable chicken-pathogenicity chromosomal islands between *Staphylococcus aureus and Staphylococcus agnetis*. 

Dear Dr. Rhoads:

I'm pleased to inform you that your manuscript has been deemed suitable for publication in PLOS ONE. Congratulations! Your manuscript is now with our production department. 

Kind regards, 

on behalf of

Dr. Feng Gao 

Academic Editor

PLOS ONE